# HAPI: An efficient Hybrid Feature Engineering-based Approach for Propaganda Identification in social media

**Akib Mohi Ud Din Khanday**[1], **Mudasir Ahmad Wani**[2]\*, **Syed Tanzeel Rabani**[3], **Qamar Rayees Khan**[4], **Ahmed A. Abd El-Latif**[2,5]

**1** Dept. of Computer Sciences & Software Engineering-CIT, United Arab Emirates University, Al Ain, United Arab Emirates, **2** EIAS Data Science Lab, College of Computer and Information Sciences, Prince Sultan University, Riyadh, Saudi Arabia, **3** Department of Computer Science, Islamic University of Science and Technology, Jammu & Kashmir, India, **4** Dept. of Computer Science, Baba Ghulam Shah Badshah University, Rajouri, Jammu & Kashmir, India, **5** Department of Mathematics and Computer Science, Faculty of Science, Menoufia University, Shebeen El-Kom, Egypt

\* mwani@psu.edu.sa

**Data Availability Statement:** Access to the data repository and source code associated with this project will be publicly accessible on our GitHub profile (https://github.com/Akibkhanday/

## Abstract

Social media platforms serve as communication tools where users freely share information regardless of its accuracy. Propaganda on these platforms refers to the dissemination of biased or deceptive information aimed at influencing public opinion, encompassing various forms such as political campaigns, fake news, and conspiracy theories. This study introduces a Hybrid Feature Engineering Approach for Propaganda Identification (HAPI), designed to detect propaganda in text-based content like news articles and social media posts. HAPI combines conventional feature engineering methods with machine learning techniques to achieve high accuracy in propaganda detection. This study is conducted on data collected from Twitter via its API, and an annotation scheme is proposed to categorize tweets into binary classes (propaganda and non-propaganda). Hybrid feature engineering entails the amalgamation of various features, including Term Frequency-Inverse Document Frequency (TF-IDF), Bag of Words (BoW), Sentimental features, and tweet length, among others. Multiple Machine Learning classifiers undergo training and evaluation utilizing the proposed methodology, leveraging a selection of 40 pertinent features identified through the hybrid feature selection technique. All the selected algorithms including Multinomial Naive Bayes (MNB), Support Vector Machine (SVM), Decision Tree (DT), and Logistic Regression (LR) achieved promising results. The SVM-based HaPi (SVM-HaPi) exhibits superior performance among traditional algorithms, achieving precision, recall, F-Measure, and overall accuracy of 0.69, 0.69, 0.69, and 69.2%, respectively. Furthermore, the proposed approach is compared to well-known existing approaches where it overperformed most of the studies on several evaluation metrics. This research contributes to the development of a comprehensive system tailored for propaganda identification in textual content. Nonetheless, the purview of propaganda detection transcends textual data alone. Deep learning algorithms like Artificial Neural Networks (ANN) offer the capability to manage multimodal data, incorporating text, images, audio, and video,

PropTweet/tree/Akibkhanday-dataset) and Kaggle repository (https://www.kaggle.com/datasets/akibkhanday/textual-propaganda) under the repository named PropTweet and Social Media Propaganda respectively.

**Funding:** The author(s) received no specific funding for this work.

**Competing interests:** The authors have declared that no competing interests exist.

thereby considering not only the content itself but also its presentation and contextual nuances during dissemination.

## Introduction

Propaganda refers to the dissemination of biased or misleading information to influence the opinions, beliefs, attitudes, or behaviors of individuals or groups in a particular direction. The word "propaganda" was coined before World War 2. The following are some early descriptions of propaganda: In 1922, Walter Lippmann's book "Public Opinion" included the first concept of propaganda. "Propaganda is an attempt to change the image to which men react, to substitute one social pattern for another" [1]. In 1928, Bernays described propaganda as a "constant, persistent attempt to produce or form events to influence people's general relations to an undertaking, thinking, or gathering" [2]. Harold D. Lasswell described propaganda in 1938 as "the technique of controlling human behavior through the manipulation of representations" [3]. There was a change in the meanings of propaganda after World War II. Jacques Ellul described propaganda as, "a variety of tactics employed by a dispersed group that needs to realize the complex or uninvolved interest in its activities of a large number of people, mentally brought together by mental controls and consolidated in an association" [4]. Garth S. Jowett and Victoria O'Donnell described it as, "the deliberate, systematic attempt to form attitudes, manipulate cognitions, and guide behavior to obtain a response that furthers the propagandist's desired" purpose" [5]. Various text analytics functions, such as sentiment analysis, are now used for political gain [6, 7]. Technological advancements have resulted in the creation of fast and accurate models for prediction and analysis [8–10]. Machine learning is showing promising results in every field of study [11].

Online Social Networks(OSNs) are described by interactions in the context of knowledge exchange, such as Texts, video files, and Uniform Resource Locator (URL) links. Individuals' cognitive decision-making processes and their relationships with others in society contribute to the spread of knowledge in OSNs. Several times in recent years, the capacity of OSNs to exploit the views of broad segments of society has been demonstrated [7, 12]. OSNs have become indispensable networking and e-commerce resources. Misinformation, fake identities, spamming, malware, and other forms of deception are popular on social media. The use of social engineering approaches to collect sensitive information to hack into computer networks is more of a concern in the cognitive domain than in the technological domain. A crucial element of cybersecurity is the ability of different systems to identify and counter such attacks. OSNs are often referred to as social computing systems. They have evolved into a vital source of personal, political, financial, health, legislative, religious, and entertainment knowledge. They've recently been exposed to a variety of semantic attacks aimed at manipulating information content and, as a result, changing user behavior. We intend to research semantic attacks in OSNs, with a focus on Twitter. OSNs are perfect channels for delivering information cost-effectively and straightforwardly. The manipulation of this media to spread negative propaganda is both useful and harmful. Close-knit networks, including professional or personal connections, can allow false propaganda to spread across the network at a tremendous pace. The media often distribute misinformation, disinformation, and propaganda. Although the repercussions are usually limited to a small percentage of online users, recent attacks have resulted in far-reaching consequences due to a coordinated effort by a group of attackers [13]. Communal attackers could successfully initiate semantic attacks by exploiting societal tensions. This

can cause fear and uncertainty in the minds of those who are affected. Even during election seasons, orchestrated AstroTurf is used to manipulate political conversations [14, 15]. The accounts that have been compromised are being used to spread disinformation, and they can also be used to spread propaganda. Propaganda is a type of covert attack that is characterized as "the systematic and deliberate process of shaping attitudes, influencing thoughts, and directing the behavior of an individual to achieve a propagandist's desired purpose".

Propaganda is primarily used to gain people's trust in an individual, group, or a political party. Since propaganda is more likely to be sent to a large audience than to a single individual, propaganda messages are more often homogeneous. Political propaganda has gained the attention of researchers all over the world. Political propaganda played a major role in Donald Trump's victory in the 2016 US presidential election [16]. Radical propaganda can be disseminated in four ways: philosophical and religious themes, crime, sectarian debate, and prominent actors and events [17]. Propaganda recognition in online social networks is a necessity of the hour. The propagandist's information may appear to be unquestionable and accurate. The propagandist intends to promote his or her goals, while there might seem to be a definite purpose and even a conclusive conclusion, the true intent is likely to be hidden. It aims to manage public opinion and exploit behavioral trends by controlling the flow of information. The term "propaganda" is sometimes used to refer to the manipulation of public opinion. Some of the novel contributions of this work are:

- A data annotation scheme is proposed for labeling the data into the binary class of propaganda and non-propaganda.

- A novel Dataset is generated based on the annotation scheme and data is extracted from Twitter using its Application Program Interface(API).

- A Machine Learning-based framework is proposed to identify propaganda and non-propaganda tweets.

- Hybrid feature engineering is proposed by merging various features like Term Frequency/ Inverse Document Frequency (TF/IDF), Bag of Words, Tweet Length, and Sentimental Features.

This proposed work introduces a novel hybrid feature selection methodology that leverages the strengths of various techniques to improve the accuracy and reliability of propaganda identification systems. By fusing the capabilities of different feature selection algorithms, our approach aims to enhance the effectiveness of propaganda detection models in diverse contexts. It is a promising approach to address the complex and evolving nature of propaganda in contemporary media. The overall structure of the article starts with the introduction of social networks and propaganda, followed by detailed literature about Misinformation, Propaganda, and Disinformation. Methodology section discuss proposed methodology in detail and the results generated through the proposed hybrid feature engineering technique are discussed in Results and Discussion section. Validation section Validates the proposed work with previous works and Finally the conclusion section concludes our work by giving future directions in current research.

## Literature review

OSNs have become critical media for the dissemination of disinformation due to users' freedom of speech, lack of filtering tools such as checking and editing available in conventional publishing, and a lack of transparency. The propagation of misinformation is aided by information asymmetry in OSNs. The proliferation of social networks deals without the use of

conventional filters such as editing. With the introduction of Web 2.0, there has been an increase in citizen journalism, resulting in faster dissemination of information through multiple online social networking channels such as blogs, emails, photo and video sharing platforms, bulletin boards, and so on. To summarise, and dissemination of various versions of information, including misinformation, disinformation, and propaganda, entails the dissemination of false or misleading information through an information diffusion mechanism, where all users may be unaware of the falsehood in the information.

The authors [18] demonstrated the use of rumors as part of a concerted marketing campaign to manage a population for the purposes of consumerism and war. The authors discuss 'rumor bombs,' which promote rumor as a privileged communication strategy with high effectiveness in managing population beliefs. They are characterized by rapid dissemination in society through electronic media. The word "misinformation" was coined to describe any kind of false information that spreads across social media. According to [19] people are willing to accept misinformation or inaccurate facts which are influenced by the previous views and opinions of influential actors. People tend to accept things that confirm their previous beliefs without challenging them. Authors in [20] studied Cognitive Psychology that backs this idea as well. According to authors, preexisting political, religious, or social views lead people to accept knowledge without verification, if it supports their beliefs. It is also difficult to counter certain ideological and personal convictions. Another interesting finding was that combating misinformation could lead to the biases being amplified and reinforce users' [21, 22] investigated political AstroTurfing in the form of meme spreading on Twitter. The research group discovered a variety of accounts sending out duplicate messages and even retweeting messages from the same few accounts in a tightly linked network while investigating political election campaigns in the United States in 2010. Similar sentiments were expressed by [23], the authors viewed rumors as potential "narrative landmines". The two aspects of narration are equally important: what is said and how it is said. Although facts are relevant, truth is all about a person's pre-existing and current understandings. The impact of misinformation spread has been extensively researched using behavioral science approaches. The study [20] contains an assessment of all facets of disinformation and the ways to correct it. The author [24] discusses the cognitive factors involved in the propagation of misinformation at the personal level, as well as the explanations for deliberate and unintentional dissemination of false facts. The internet has resulted in the fragmentation of the information landscape. [25] states that the idea of people being selectively exposed to knowledge sources that support their beliefs become very common. As a result, Cyber-ghettos or echo chambers have emerged, where ties in social networks follow like-minded people with similar viewpoints.

The information about an incident as it unfolds, such as casualty estimates in a natural disaster, is rarely reliable at first, and the figures are revised or modified over time. Despite the fact that the media is one of the most significant sources of misinformation, the dissemination of misinformation is sometimes seen as innocuous. Governments and lobbyists, political interests, rumors, and works of fiction are all essential sources of disinformation [20]. The researchers [26] describe cognitive hacking and its numerous countermeasures. The authors define a cognitive hack as one that alters a user's expectations and, as a result, their behavior. The amount of time between the dissemination of false information on the internet and the resulting shift in user behavior is critical. As covert attacks have shown, detecting such attacks, as well as providing effective steps to deter cognitive hacking, is a difficult task.

The authors [27] enumerated a variety of potential Internet instances of misinformation. Total, out-of-date, and biased content, pranks, inconsistencies, incorrectly translated data, software incompatibilities, unauthorized revisions, factual errors, and scholarly misconduct are just a few examples. However, with the introduction of Web 2.0, the list has exploded, and

social media is now considered one of the most important sources of knowledge, including disinformation. The internet functions like a post-modern Pandora's box, revealing a plethora of information-related claims that are difficult to dismiss [22]. The authors [28] created a method called "Seriously Rapid Source Review" (SRSR) for trained journalists to use in filtering and evaluating the veracity of social media sources. To validate information sources, the device employs a variety of filtering and information cues, including content-based functionality, aggregation, and location of data. Ordinary users would have limited use for the device in determining the reliability of sources. According to [29] the authors perceived source reliability and cognitive elaboration are critical factors in determining the trustworthiness of sources. Cognitive elaboration entails active involvement in information processing and can include tasks such as content discussion. For knowledge verification, the work clearly shows the value of source reliability and recent changes. [30] suggested a system for detecting and tracking political violence in social media. They looked at Astroturf political campaigns on micro-blogging sites, which include politically motivated individuals and organizations using numerous centrally managed accounts to give the impression of widespread support for a candidate or viewpoint. [31] studied the reliability of Twitter in serious circumstances. The spread of speculation on Twitter differs from the spread of reliable news, according to an examination of tweets linked to the earthquake in Chile in 2010. Rumors are more likely to be challenged. The authors manually picked verified news and rumors from a collection of tweets following the earthquake to examine trends of knowledge dissemination in the form of retweets in the network. With the aid of this report, the use of Twitter as a collaborative filter mechanism was demonstrated. The authors also confirmed the accuracy of using aggregate analysis of tweets to identify rumors. The researchers [32] investigated the credibility of tweets during high-impact events. To assess the authenticity of tweets and their sources, the authors used source-based and content-based features. Several words, special symbols, hashtags, pronouns, URLs, and metadata including retweets were all used as content-based features in the tweets. The reputation of a user was assessed using source-based factors such as the number of followers, followers, and age [33]. RankSVM and Relevance feedback algorithms were used to assess the features for credibility [34]. The necessity to define ground truth using human annotation was a limitation of their work.

The researchers [35] studied the structure of social networks during the election campaign. They develop a model for Opinion Dynamics that occur during an election campaign by demonstrating that there is a large region in the phase diagram where two antagonistic parties survive by garnering a finite fraction of the votes, implying that pluralism in the electoral system exists. [36] analyzed and mined data from online social networks about civil unrest. They extracted data from the real event, identified contributing factors, and analyzed the event's evolution. The underlying complex mechanisms and characteristics of political entrepreneurship can be investigated in this study. Researchers in study [37] investigated the impact of social bots on politics (Political propaganda through Social Bots). They discovered that social bots play a critical role in the dissemination of fake news, and accounts that consistently spread disinformation are much more likely to be bots. If the user's age and position can be determined from their social networking sites, the research can be expanded on Bots. [38] proposed a semantic graph-based approach for radicalization detection in Social media. They discovered that pro-ISIS users talk about religion, historical events, and race, while anti-ISIS users talk about politics, geographical locations, and counter-ISIS interventions. The research may be used to evaluate Arabic tweets in the future and examine the applicability and success of their semantic functionality on other social media sites, such as Facebook and Instagram. They have posted shady details on social media. The [39] identified propagandistic communities based on leader ranker and network constraint algorithms. The researchers [40] identified the

socially mediated type of populist communication profoundly affected by the specific nature of social media. Based on their assessment the nationalist communicative ideology appeals to the people, attacks on the elite, and exclusion of others. According to the findings, at least one of the three dimensions of populist ideology can be found in 67% of the articles. [41] compared two models for Terrorist Group Detection: GDM (Group Detection Model) and OGDM (Offender Group Representation Model). The results revealed that Detecting an offender group, a terrorist network or even a part of a group (subgroup) is also important and valuable. [42] Detected and Tracked Political Abuse in Social Media. They looked at AstroTurf political campaigns on micro-blogging sites, which include politically motivated individuals and organizations creating the impression of widespread support for a candidate or viewpoint by using multiple centrally managed accounts. They described a machine learning system for detecting the early stages of the viral spreading of political disinformation on Twitter that incorporates topological, content-based, and crowd-sourced features of knowledge diffusion networks. They came close to a 96% accuracy rate. The authors used the Twitter API to retrieve data from Twitter. Hashtags, URLs, and memes. Crowd-sourcing the annotation of truthy memes may be explored further in the future. These researchers were unable to collect sufficient crowd-sourcing data (only 304 'truthy' button clicks, which were mostly associated with meme popularity), but these annotations may be useful with more data. [43] discussed Evolving Strategies in Defense and Intelligence Propaganda. The authors analyzed the changing Anglo-American counter-terror propaganda tactics that spanned the Afghanistan and Iraq wars, as well as the construction industry. While a thorough examination of British and American documentary sources was carried out, the primary method of data collection was exploratory elite face-to-face and telephone interviews. The data was subjected to thematic analysis, which identified implicit and explicit themes or ideas within the data and coded them for analysis.

The study in [44] compared classification results obtained from expert and amateur annotations, it was seen how annotator knowledge of hate speech affects classification models. They have annotations for hate speech on 6,909 tweets by Crowd-Flower annotators and annotators with theoretical and applied knowledge of hate speech, referred to as amateur and expert annotators. Expert annotations, they discovered, can generate models that work similarly to previous classification efforts. It can be investigated further in terms of socio-linguistic characteristics such as gender and position.

Authors in [45] performed automated hate speech detection and the problem of offensive language. Hate speech is described as a language that communicates hatred toward a specific group or is intended to be negative, humiliating, or insulting to its members. They gathered tweets containing hate speech keywords using a crowd-sourced hate speech lexicon. They used crowd-sourcing to categorize a subset of these tweets into three groups: those containing hate speech, offensive words, and none. To differentiate between these various groups, they trained a multi-class classifier. Overall precision was 0.91, the recall was 0.90, and the F1 score was 0.90 for the highest-performing model. Almost 40% of hate speech is misclassified: the hate class's accuracy and recall scores are 0.44 and 0.61, respectively. About 5% offensive tweets and 2% of harmless tweets have been incorrectly labeled as hate speech. People who use hate speech should be studied more closely, with an emphasis on both their individual characteristics and motives, as well as the social systems in which they are rooted. Table 1 shows some of the recent studies in the area of propaganda and text mining.

The Following conclusions can be inferred from the literature:

- Majority of work is done on News Articles.

- Data can be extracted from social networks to check whether it is propaganda or non-propaganda.

**Table 1. Recent work related to propaganda.**

| Reference | Year | Technique | F-Measure | Precison | Recall |
|---|---|---|---|---|---|
| [46] | 2020 | Part of Speech | 51.55 | 56.54 | 47.37 |
| [47] | 2020 | Conditional random Fields | 49.15 | 59.95 | 41.65 |
| [48] | 2020 | Embeddings | 49.10 | 53.23 | 45.56 |
| [49] | 2020 | Bag of Words | 47.66 | 50.97 | 44.76 |
| [50] | 2020 | Embeddings | 46.6 | 58.61 | 37.94 |
| [51] | 2020 | n-Grams | 44.68 | 55.62 | 37.34 |
| [52] | 2020 | Part of Speech | 43.86 | 42.16 | 45.7 |
| [53] | 2020 | Embeddings from Language Model | 43.60 | 49.86 | 38.74 |
| [54] | 2020 | Part of Speech | 42.21 | 46.52 | 38.63 |
| [55] | 2020 | Bag of Words | 33.21 | 24.49 | 51.57 |
| [56] | 2020 | Embeddings | 23.47 | 22.63 | 24.38 |
| [57] | 2020 | Embeddings | 18.18 | 34.14 | 12.39 |

- Since propaganda is a Multimodal classification task only textual data is being investigated more.

- Feature engineering can be explored in order to improve the accuracy of Machine Learning and Deep Learning Algorithms.

News article analysis has been the main focus of most research in this area. On the other hand people in the modern era share their thoughts and viewpoints on social media platforms, and there is a great potential to use online social media data for propaganda detection. These platforms are excellent resources for current events and provide insightful information about the propagation of propaganda and its impact on public opinion. The performance of several machine learning classifiers in this context has not been sufficiently enhanced by feature selection techniques. By determining the most relevant and discriminative features in the dataset, feature selection will play a critical role in managing the effectiveness of classifiers. Moreover, the methods developed in the past can be useful for enhancing the propaganda identification systems. For example, the studies [58] and [59] proposed citation intent classification approach and the Arabic sentiment analysis mechanism respectively, demonstrates the effectiveness of word embeddings and deep learning techniques, which can be adapted to enhance context understanding and detection accuracy in our propaganda identification model. Similarly, the authors in [60] introduces a sophisticated MPAN model with multilevel parallel attention mechanisms, which can improve precision and recall by focusing on relevant parts of the text. Additionally, the MPAN model's robustness across different domains highlights its potential to handle the diverse nature of propaganda across various social media platforms, making our approach more resilient and effective.

## Methodology

OSNs have become the tools for communication and a massive amount of information is being disseminated through them. Propaganda is a form of information that is defined as "a form of communication that attempts to achieve a response that furthers the desired intent of the propagandist" [5]. A methodology is being proposed to identify the propaganda on social networks and is depicted in Fig 1.

The major phases of our methodology are Data Extraction, Data Preprocessing, Feature Extraction, and Machine Learning classifiers. The model developed based on the proposed

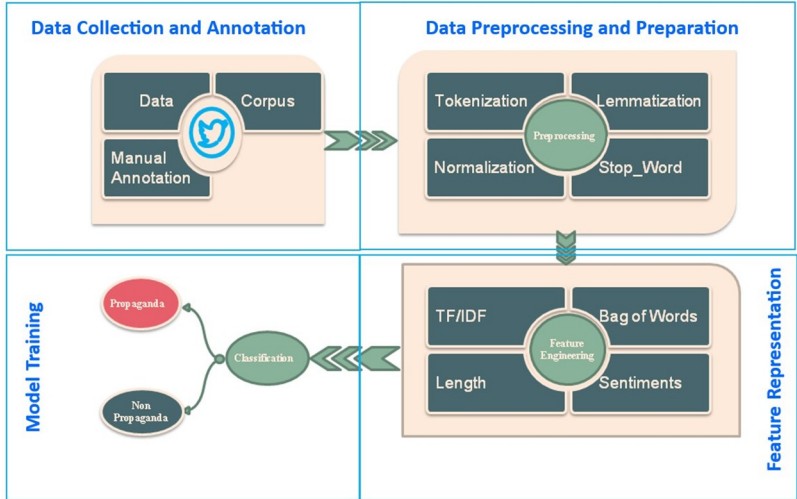

**Fig 1. Proposed framework for propaganda identification.**

methodology has been named the Hybrid feature engineering Approach for Propaganda Identification(HaPi). The flowchart of the processes is shown in Fig 2.

## Data collection

Data is the main component in every research. Due to the nonavailability of a dataset regarding propaganda, we generated a novel dataset for our research work. In our work, data is extracted from online social network platforms using various methods such as APIs, Crawlers, etc. In this work, we extracted data from Twitter, a social networking site using its API. Twitter is a real-time, highly social micro-blogging site that allows users to publish short status updates known as tweets, which are displayed on timelines. In their 280-character content, tweets may contain one or more entities, as well as references to one or more places in the actual world. It is a public platform with millions of tweets published every day from millions of user accounts, all of which have academic and commercial value. The Twitter API allows developers to access Twitter's data. With successful usage of its API, an understanding of users, tweets, and timelines is very important. In Twitter, there are three types of API: Search API, Streaming API, and REST API. The Twitter Search API gives Twitter developers access to data containing specified keywords, tweets mentioning a certain person, and tweets from that user. The Search API prioritizes relevance, and some data is missing using this API. The Streaming API delivers a dataset that is relatively more complete. Twitter's Streaming API gives real-time data in massive amounts to Twitter developer apps. Twitter developers must create and maintain a long-lived HTTP connection to use the Streaming API. Researchers use the Streaming API to collect data linked to keywords or geo-tagged tweets from a certain region by setting up various parameters. The most significant drawback of the Streaming API is that it only delivers a sampling of real-time data. The REST API gives developers access to the data of a specific person, such as timelines, status updates, and other details. Unlike the Streaming API, which delivers real-time data, the REST API gives a user's past data. Developers can also utilize the REST API to interact with the site by publishing tweets, following other users, and editing user profiles. Many programming languages, including R and Python, are used to collect, process, and analyze Twitter data. For getting access to Twitter's API we need to go through different phases.

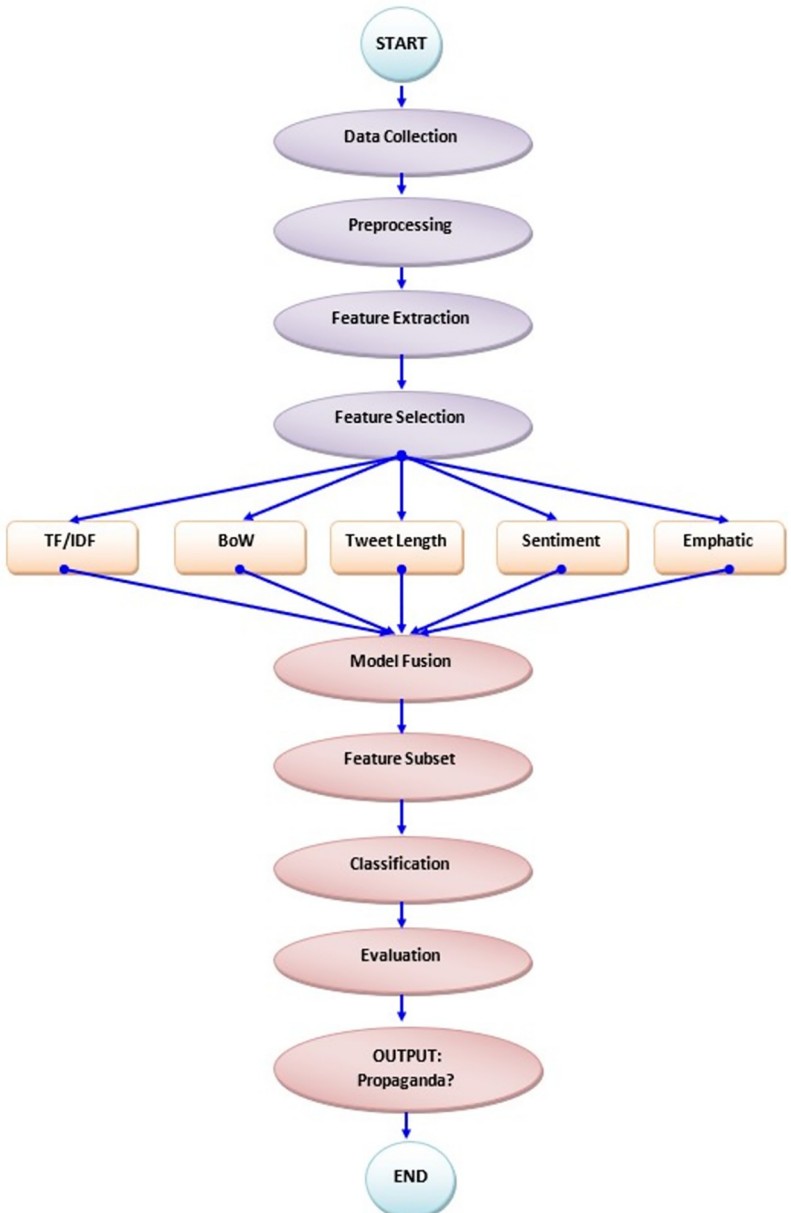

**Fig 2. Overall flowchart of the process for identifying propaganda.**

The framework for data extraction shown in Fig 3 depicts the various phases by which we get access to Twitter's API.

The process captions are numbered sequentially from steps 1 to 6, depicting the flow of the procedure. In the initial step, the user sends their credentials to the Authorization server, where authentication occurs, and an authentication code is issued. This authentication code is subsequently transmitted to the web client. The web client, in turn, forwards the credentials to the Server and acquires Access Token keys. These Access Token keys are then passed on to the end user, who utilizes them for extracting data from Twitter.

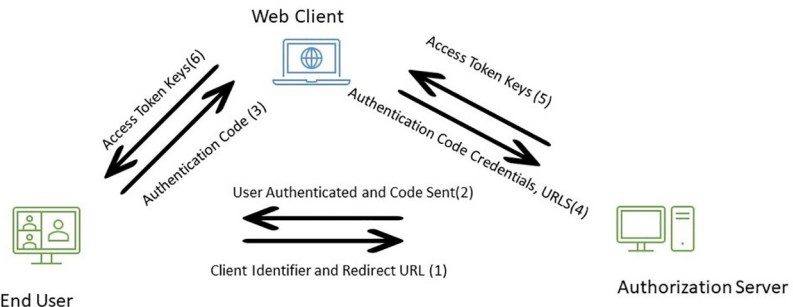

**Fig 3. Framework for data extraction from Twitter.**

Twitter has a large user base with millions of active users posting tweets daily. This popularity makes Twitter a valuable source for studying and analyzing propaganda. Handling data from multiple platforms can become resource-intensive and complex. Focusing on one platform allows for a deeper exploration of platform-specific propaganda phenomena.

Data is extracted based on some events using various hashtags and keywords. A keyword-based search is being run to extract the data. Some of the hashtags are #Propaganda #Hoaxes #Falseflag etc. In our work, we have focused on events regarding Politics such as elections, government schemes, etc. From May 2020 to May 2021, approximately 1 million tweets were extracted using these hashtags. For crawling data from Twitter, the R packages twitteR, devtools, Rcurls, igraph, RoAuth, thttr, and base64encr are used. R3.5.1, a statistical program, is used to collect data from the Twitter network in real time. The collected dataset (https://github.com/Akibkhanday/) has 16 features including tweet text, time, date created, screen name, etc.

The selected keywords were carefully chosen to target propaganda-related content, it is challenging to create an exhaustive list of all possible keywords and variations that propagandists might use. Propaganda content can be diverse and may employ subtle or coded language that does not match the selected keywords and those that were not considered in the study. The chosen keywords were designed to capture a significant portion of propaganda-related content. Due to the limitations of keyword-based collection, there is a possibility of missing some instances of propaganda that do not include the specified keywords. These missed instances may represent a subset of propaganda-related content that is not directly identifiable using keyword matching. The dataset has been extracted in line with Twitter's data accessibility policy in 2020. For privacy concerns, the identification numbers and screen names associated with the Twitter accounts have been removed from the dataset. This modified dataset is made public in the data repository, which can be accessed in our repository (https://github.com/Akibkhanday/PropTweet/tree/Akibkhanday-dataset). Due to the substantial size of the dataset, it has been divided into 10 parts, labeled from part1 to part10. This division facilitates easier handling and processing of the dataset. Each part of the dataset contains a subset of the overall data, organized in a manner that allows for efficient retrieval and analysis. By removing identification numbers and screen names, the dataset addresses privacy concerns and ensures compliance with Twitter's policies regarding data usage and user privacy. This approach enables researchers and analysts to access and analyze the dataset while respecting the privacy rights of Twitter users. Also, implementation procedure and code have also been made public in the repository(https://github.com/Akibkhanday/PropTweet). This inclusion ensures that not only the data but also the methodology used to process and analyze it are openly accessible to researchers.

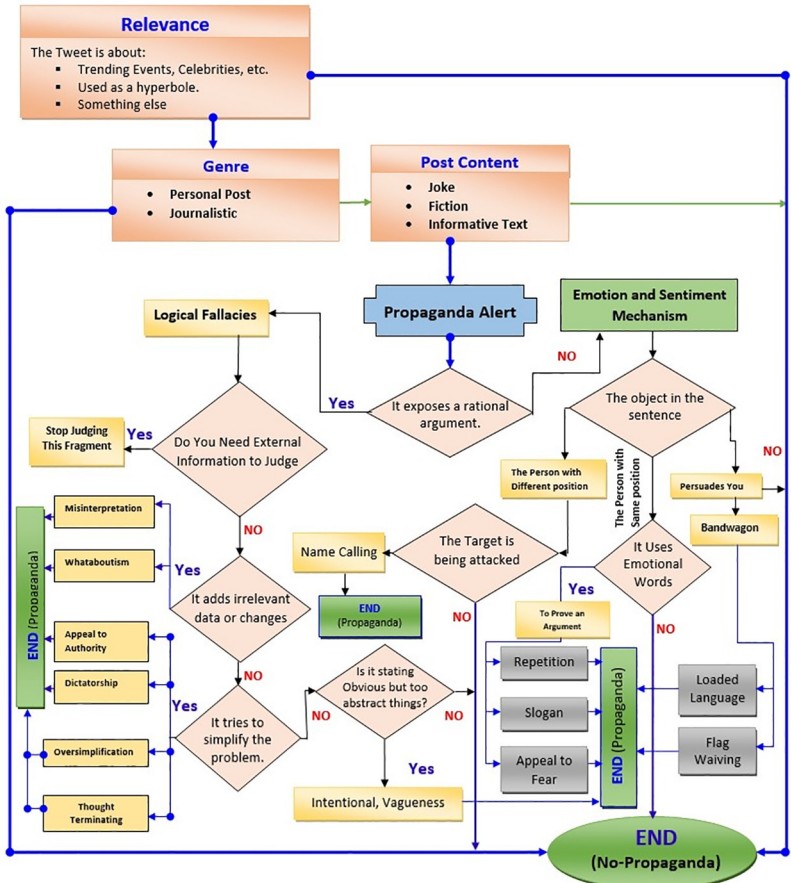

**Fig 4. Proposed annotation scheme using various propaganda techniques.**

**Data annotation.** Annotation of data is performed manually so that supervised machine learning algorithms can be trained and tested. We proposed an annotation scheme framework for labeling a tweet into binary classes i.e., Propaganda and Non-Propaganda which is depicted in Fig 4.

Following the extraction of data from Twitter, the dataset underwent annotation based on the proposed annotation scheme. Various propaganda-spreading techniques, as identified by researchers, were investigated to formulate this annotation scheme. Subsequently, 14 distinct techniques were integrated into the scheme to classify or label tweets as either propaganda or non-propaganda.

The reliability of the labeling process was assessed through an examination of agreement among annotators. A representative sample of 500 tweets was selected from the overall corpus for inclusion in the study. Three annotators were assigned the task of non-hierarchically labeling the posts based on diverse propaganda strategies. Out of the selected tweets, 38 instances resulted in disagreement among annotators and were consequently excluded from the corpus. Inter-annotator agreement was evaluated using the Kappa coefficient for the various tweets.

Post the formulation of the annotation scheme, it was determined that the scheme exhibited exceptional reliability in the annotation of tweets. Additionally, the dataset demonstrated a balanced distribution of data for both propaganda and non-propaganda categories. Following annotation, the length of tweets was assessed, considering both word count and character

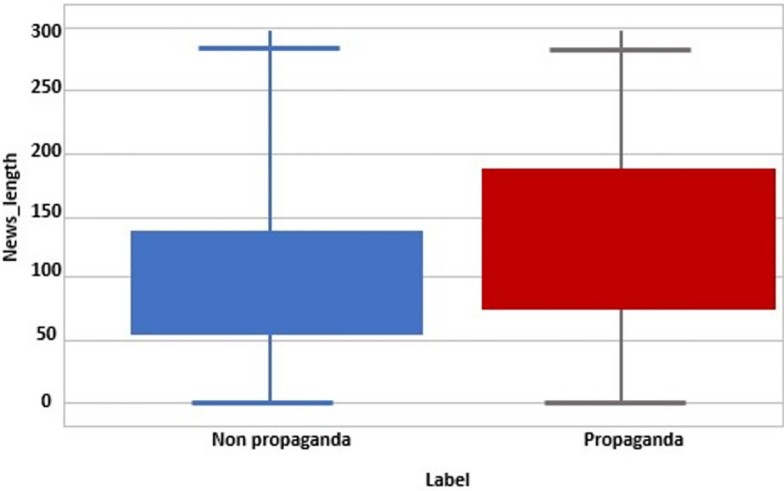

**Fig 5. Labeled dataset with their corresponding lengths.**

count. The analysis revealed that propagandistic tweets tend to be longer than their non-propagandistic counterparts. Fig 5 shows labeled datasets with their character length.

To calculate the Inter Annotator Agreement for classifying tweets into "propaganda" and "non-propaganda" classes, we use several Cohen's Kappa. Cohen's Kappa measures the agreement between two annotators while taking into account the possibility of agreement occurring by chance. It provides a value between -1 and 1, where:

*Kappa* < 0: Less agreement than expected by chance. *Kappa* = 0: Agreement equal to that expected by chance. *Kappa* > 0: Agreement better than expected by chance.

Following exploratory analysis, word clouds for propaganda and non-propaganda text were generated using the Wordcloud library in Python, illustrated in Figs 6 and 7.

Upon close examination, it is evident that posts with propagandistic content tend to revolve around particular subjects, such as politics, religion, trending events, and influential people. This finding is corroborated by a detailed examination of the word cloud linked to propaganda, which shows that terms of politics, religion, ISIS, and Islam appear most frequently. This implies a deliberate attempt to advance specific ideologies or agendas by disseminating inaccurate or biased information on various digital platforms.

## Data preprocessing

The data extracted from online social networks is usually noisy. Since the noisy data can't be supplied to the machine learning algorithm, for training and testing it needs to be refined. Preprocessing is critical for deciphering the meaning of brief texts in classification applications. It significantly impacts total system performance, but it has received less attention in the literature than feature extraction and classification. Preprocessing tweets includes preparing them for additional tasks such as event recognition, bogus information detection, sentiment analysis, etc.

People generally adhere to their own set of informal language rules on social media. As a result, each Twitter user has a writing style, including abbreviations, non-standard punctuation, and misspelled words. Emoticons and emojis are used in tweets to convey complexity, sentiment, and ideas. Slang and acronyms are common in tweets, URLs, hashtags, and user

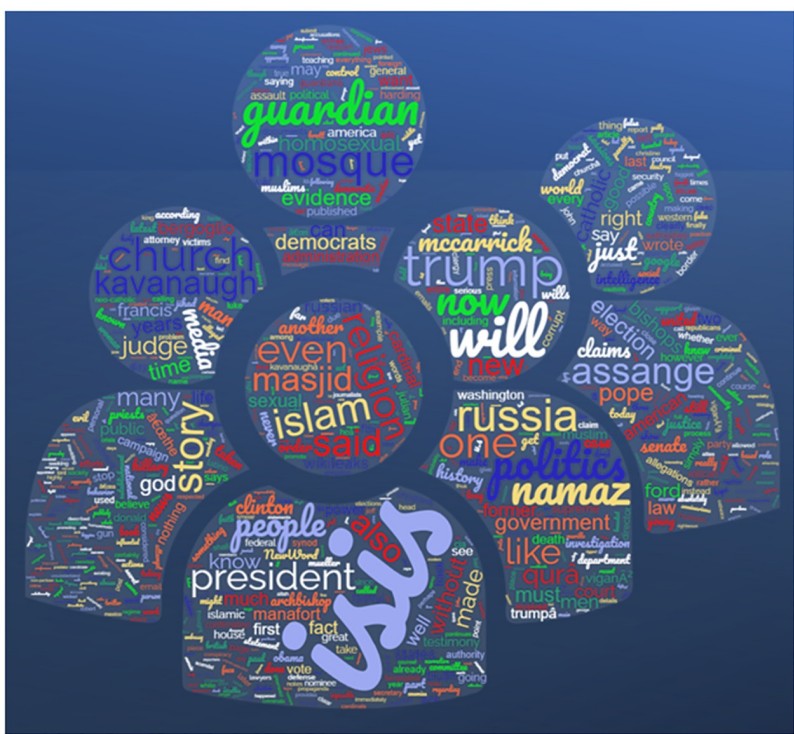

**Fig 6. Wordcloud of propaganda text.**

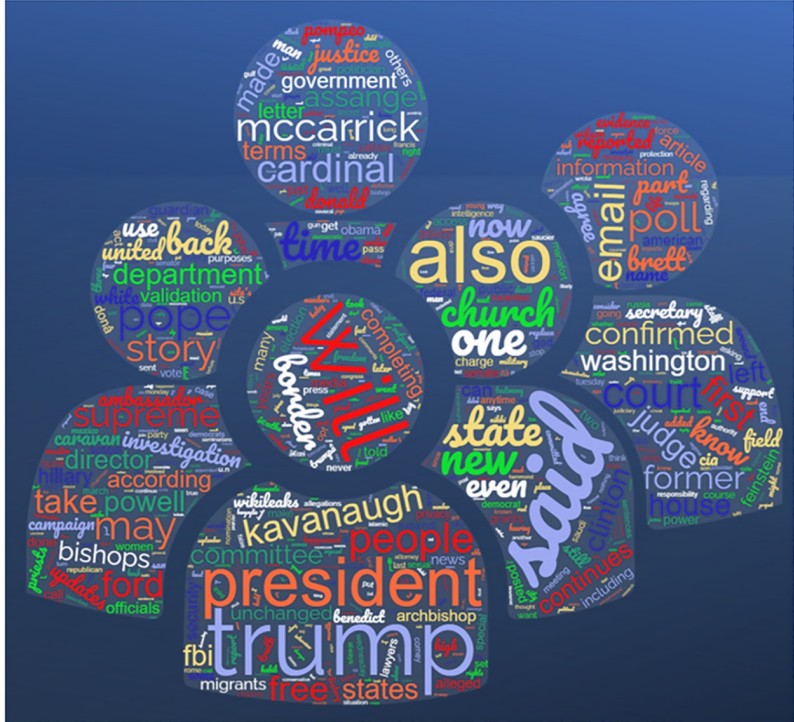

**Fig 7. Wordcloud of non-propaganda tex.**

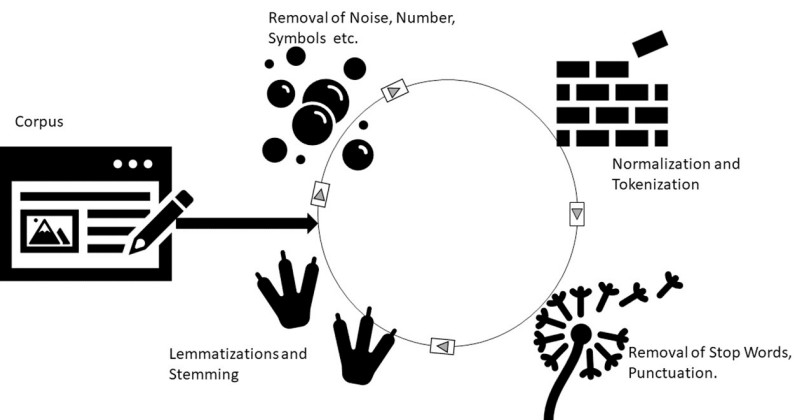

**Fig 8. Pictorial representation of various steps in data preprocessing.**

mentions. For refining the dataset we used different techniques like Tokenisation, normalization, Punctuation removal, and Lemmatization, etc. The pictorial representation of major steps in data preprocessing is shown in Fig 8.

Following are the different techniques that are used for performing data preprocessing:

- **Tokenization**: in this phase, the tweets are divided into tokens such that we can easily refine the text by removing unwanted words, digits, etc.

- **Normalization**: In this task, all the tokenized words are converted into lowercase also the whitespaces are removed.

- **Removal of Noise, URLs, Hashtags and User Mentions**: Unwanted strings and Unicode are considered to be the byproducts of the crawling process and they add noise to the data. Additionally, tweets include URLs (that link to extra information), user mentions (@username), and the hashtag symbol (#propagandahoaxes) for connecting the message to a specific subject or to express the mood. These clues provide additional information that is important to humans, but they do not give any information to machines and can be regarded as noise. All user mentions, URLs, Hashtags, and Noise are removed in this step.

- **Word Segmentation**: Word segmentation is performed such that the content of the text is extracted. The technique of separating the phrases/content/keywords used in a hashtag is word segmentation. For example, #sometrendingtopic is segmented into three words: some, trending, and topic. This phase can assist machines easily in recognizing and identifying the content of tweets without the need for human interaction.

- **Replacing Emoticons and Emojis**: Twitter users replace emoticons and emojis with a variety of symbols such as":-)", ";-)",":-(", etc. to express sentiment or opinion. To effectively classify tweets, it is also necessary to record this crucial information. These expressions and emoticons were replaced with their respective word meanings, for example, ":-)" is replaced with "happy" and ":-(" with "sad".

- **Abbreviations and slang**: Twitter's character limit restricts natural language usage, encouraging users to employ acronyms, abbreviated phrases, and slang in their online posts. An abbreviation is an abbreviation or acronym of a term, such as MIA (missing in

action), gr8 (great), ofc (of course), and so on. Slang is also an informal style of expressing thoughts or meaning that is sometimes limited to specific individuals or environments and is deemed casual. "OMG" seldom ever refers to the literal expansion of "oh my God", but rather to a surprise or emphasis. As a result, it's critical to deal with such casual insertions in tweets by replacing them with their true word meaning, which leads to improved automatic classifier performance without information loss. Abbreviations and slang were translated into word meanings, which were then easily understood using normal text analysis methods.

- **Punctuation Removal**: Punctuation is used by social media users to portray feeling and emotion, which is easily understood by humans but not as effective for the automated classification of short texts. As a result, removing punctuation while preprocessing text is good for automated classification tasks like sentiment analysis which is typical to practice. However, some punctuation characters, such as'!' and '?', can be used to convey emotion. Substituting a question mark or exclamation with appropriate tags, such as '!', can often indicate astonishment.

- **Stopword Removal**: The uninformative and distracting terms are usually called stop words, these mainly include conjunctions and prepositions. Stop word lists must be used with caution; while they lessen the likelihood of accumulating unnecessary data, they also risk discarding/affecting good data. In this phase, we removed all stopwords, using the English Stopword dictionary, due to their least role in the classification task.

- **Stemming**: Each word stems from its root word such that the actual meaning of the word can be generated.

- **Lemmatization**: In this phase, vocabulary and morphological analysis of words is done. Lemmatization aims to remove inflectional endings only and to return the base or dictionary form of a word, which is known as the lemma. We use spaCy's lemmatizer to obtain the lemma, or base form, of the words.

After using the above techniques data is refined and also non-English tweets are discarded from the corpus.

## Feature engineering

Feature extraction is one of the important phases in every classification task. Features are sample qualities that can be detected by machines and are relevant to a Machine Learning task on the corpus. In this phase, relevant features regarding some event are extracted, for sampling the ideology being used in the tweet, such as extremist ideology or soft ideology, etc. The proposed method introduces innovative strategies for combining different feature selection techniques. The proposed hybrid approach has the potential to be more robust against emerging propaganda tactics, given its diversified feature selection techniques. The proposed method adapts, combines, fine-tunes, and optimizes existing feature selection techniques specifically for the challenging task of propaganda detection. While the individual methods are known, their integration and application in this context can lead to novel insights, improved performance, and enhanced adaptability, making it a valuable contribution to the field of information warfare analysis. The feature selection techniques that are used for performing this task are as follows:

**TF/IDF.** Term Frequency/Inverse Document Frequency mirrors a word's significance in a tweet or in an entire corpus by giving its numerical insights. The following Eqs 1 and 2 are

used for calculating term frequency and Inverse Document Frequency.

$$TFIDF(t, w, D) = TF(t, w) * IDF(t, D) \tag{1}$$

$$IDF(t, D) = \log \frac{|D|}{1 + |(w \in D : t \in w)|} \tag{2}$$

Where $t$ is the term as a component, $w$ signifies each tweet in the corpus and $D$ is the total number of tweets in the dataset (Document space).

**Bag of Words.**   It is a way of extracting features from text and then used for training and testing of a machine learning algorithm. Bag-of-Words is a representation of text that describes the occurrence of words within a document. It involves two things:

- A vocabulary of known words.

- A measure of the presence of known words.

It is called a "bag" of words because any information about the order or structure of words in the document is discarded. The model is just concerned about the occurrence of the word in the document, and not where it is in the document. The approach is very simple and flexible and can be used in a myriad of ways for extracting features from documents. It consists of words and lemma unigrams, bigrams, and trigrams. We included bigram, and trigram words as a feature for the machine learning classifier such that more information can be extracted from the text.

**Sentiment analysis.**   In this phase, sentiments are extracted from the tweets by giving different sentiment scores to different words. We classify the tweets according to their sentiments as positive, negative, and neutral. Since this is also a classification task, we used a lexicon-based approach for performing sentiment analysis. After using the dictionary/lexicon-based approach it was found that the propagandistic tweets have mostly neutral sentiments, whereas non-propagandistic tweets have positive as well as negative sentiments in the majority of tweets. This feature is also being used for training and testing a machine-learning model.

**Tweet length.**   After performing exploratory analysis, it was found that the sentences containing propagandist messages are more likely to be longer than the non-propagandist messages. If a textual tweet has less than eight tokens it belongs to a short document otherwise it is a long document. Discrete features like Text Length (No. of characters in a sentence) and word count are included in the classification task of propaganda.

**Emphatic features.**   Many techniques are being used for propaganda such as Slogans, Name-calling, and Loaded Language, which contains Emphatic content i.e., every word begins with a capital letter or is in double quotes. Our Approach includes all these features. To identify the Emphatic word, we used a feature called "Emphatic", for a sentence whose value is set to 1 if the content is in double quotes or there is an upper letter sentence. Table 3 shows some of the selected features.

**Statistical analysis.**   The statistical results of some features are shown in Table 2. The statistical analysis is performed such that we can get much knowledge about the data that is used for classification purposes. Table 3 shows the selected vocabulary features out of all the vocabulary sets extracted from the collected data. It was found that the features selected for classification are normally distributed.

The propaganda detection tasks suffer from data sparsity, where there are limited examples of propaganda texts compared to non-propaganda texts. Hybrid features help mitigate this issue by providing additional information sources that may be more readily available, such as metadata or syntactic patterns. The components of this new method may individually be

**Table 2. Feature statistical analysis.**

| Feature | Standard Deviation | Mean | Min | Max |
|---|---|---|---|---|
| Cardinal | 0.029656995 | 0.14257 | 0 | 1 |
| Ford | 0.018193369 | 0.11460 | 0 | 1 |
| Continue | 0.013808926 | 0.09728 | 0 | 1 |
| Come | 0.024063467 | 0.13523 | 0 | 1 |
| Say | 0.016012491 | 0.10581 | 0 | 1 |
| President | 0.017218344 | 0.11264 | 0 | 1 |
| Trump | 0.023046996 | 0.12755 | 0 | 1 |

**Table 3. Features chosen based on hybrid feature engineering.**

| cardinal | president | vote | new | time |
|---|---|---|---|---|
| ford | continue | came | ray | pope |
| take | trump | judge | many | first |
| report | people | church | email | guardian |
| make | even | may | court | arrange |

known, its novelty stems from the unique way they are combined and adapted to address the complex and ever-evolving challenge of detecting propaganda within the constraints of Twitter. Also, propaganda often relies on specific linguistic cues and rhetorical techniques, such as emotional language, persuasive appeals, and manipulation of information. Linguistic features, including sentiment analysis and sentiment polarity, can help capture these cues. These hybrid features provide complementary information, which helps in improving the model's ability to discriminate between propaganda and non-propaganda texts. This is where the hybrid approach comes into play. Instead of directly selecting features these hybrid features (TF/IDF, Bag of Words, Sentimental, Tweet Length, and Emphatic) are employed in parallel. This innovative approach offers an opportunity to advance the field of propaganda detection and ensure the resilience of online discourse against manipulative tactics.

## Results and dicussion

The principal objective of this research is to formulate a classifier for categorizing tweets into two distinct classes: propaganda and non-propaganda. Following the annotation of the dataset, the feature selection techniques elucidated in the preceding sections are employed to extract pertinent features. These features are subsequently utilized as inputs for various fine-tuned machine-learning algorithms, namely Logistic Regression, Multinomial Naive Bayes, Support Vector Machine, and Decision Tree.

The choice of these algorithms signifies a comprehensive strategy encompassing a spectrum of complexity levels and interpretability. Logistic Regression and Decision Trees offer transparency, aiding in the understanding of the model's decision-making processes. Multinomial Naive Bayes is specifically tailored for text data, while Support Vector Machine (SVM) is acknowledged for its robustness. This diverse array of algorithms enables us to assess which method is most efficacious for the unique features associated with propaganda identification within our dataset.

Moreover, the selection of these algorithms is underpinned by a strategic consideration of their appropriateness within the context of our study. This involves a thorough examination of

the dataset's characteristics, the distinctive attributes of propaganda, and the overarching objectives of the problem at hand.

The Algorithm 1 HAPI (Hybrid Feature Engineering Approach for Propaganda Identification) represents our proposed approach that is used to classify and differentiate propaganda tweets from non-propaganda tweets. The labeled dataset is split in the ratio of 70:30 i.e., 70% of data is used for training the classifier and 30% used for testing.

**Algorithm 1** HAPI: **H**ybrid Feature Engineering **A**pproach for **P**ropaganda **I**dentification

```
Require: Filtered Tweets (T_input), Classifier_Name, Hyperparameters
Ensure: Propaganda Tweet (T_Pr) and Non-Propaganda Tweet (T_Np)
 1: Tokenization → T, StopWordRemoval → SW, Stemming → S,Total Number
    of Tweets → n
 2: Term Frequency/Inverse Document Frequency → TF/IDF, Bag of Words →
    B, Sentiment Analysis → Sent
 3: START
 4: for i from 1 to n do
 5:   C[i] = T_input[i] + Label //Manual Annotation
 6:   Text.csv = C[i] // Saving into .csv file
 7: end for
 8: Pro = T(Text.csv)
 9: Pro = SW(Pro)
10: Pro = S(Pro)
11: Processed.csv = Pro //Saving Preprocessed file as CSV
12: for i from 1 to n do
13:   Sent[i] = Sent(Pro[i] //Sentiment Analysis
14:   Length[i] = nchar(Pro[i]) //Lenth of Tweets
15: end for
16: Features = B(Processed, n = 1 : 3)//Upto Trigrams
17: Document Feature Matrix = Document Feature Matrix(Features)
18: Trimmed_dfm = dfm_trim(dfm,min_docfreq,min_termfreq)
19: Final_Features = Trimmed_dfm.Tfidf //Extracting TFIDF Features
20: Optimal_Features = Final_Features + Length + Sent
21: CLASSIFIER(Classifier_Name, Hyperparameters, Cross_Valid = 5,
    Optimal_Features)
22: END
```

## Logistic regression

Logistic Regression is one of the simplest supervised Machine Learning algorithms. In this work, we used Binomial Logistic Regression, where the target variable is only of two possible types: propaganda and Non-propaganda. Due to the relationship between the label and the class, Logistic Regression predicts a numerical value for the class. In general, the method determines the likelihood of a person belonging to a specific class. Pseudocode 2. represents the parameters for Logistic Regression based HaPi(LR-HaPi). After implementing the pseudocode the confusion matrix is shown in Fig 9.

## Multinomial Naive Bayes

Multinomial Naive Bayes is a statistical method that calculates the class probabilities of a given text. It can be used to tackle problems involving categorization into binary and multi-class categories. The fundamental notion is that each feature should be treated independently of the others. The Naive Bayes technique, which is based on the Bayes Theorem, calculates the probability of each attribute separately, regardless of whether or not there are any correlations between them. Pseudocode 2. represents the parameters for Multinomial Naive Bayes-based

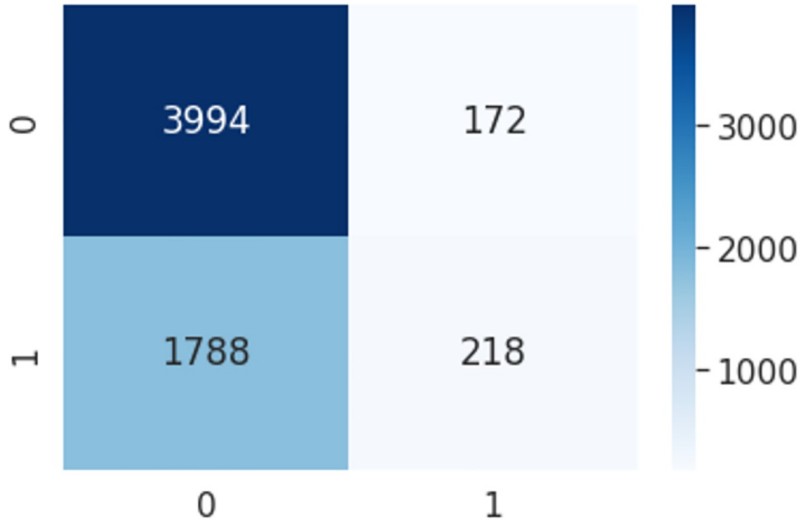

**Fig 9. Confusion metrics using LR-HaPi algorithm.**

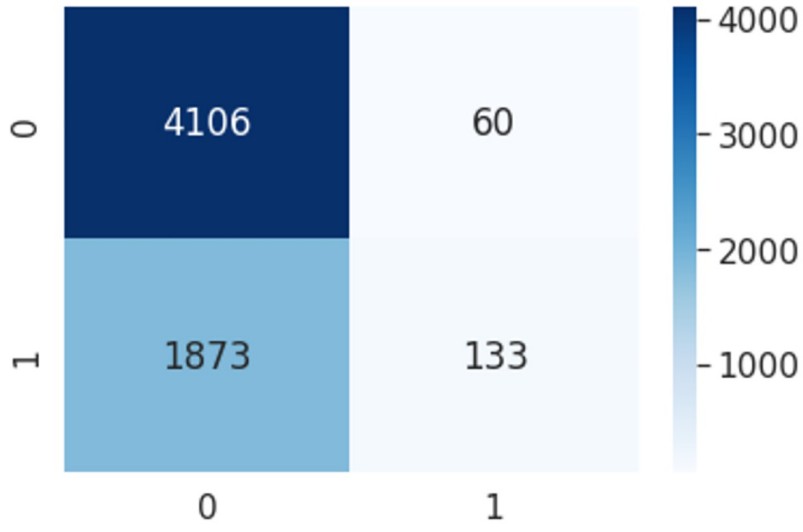

**Fig 10. Confusion metrics using MNB-HaPi algorithm.**

HaPi(MNB-HaPi). After implementing the pseudo-code the confusion matrix is shown in Fig 10.

**Algorithm 2 Parameter Tunning for Propaganda Identification**(Model Configuration)

1: **LRHyperparameters**→ C = 1.0, classweight = None, dual = False, fit-intercept = True, intercept-scaling = 1, max-iter = 100, multiclass='warn', n_jobs = None, penalty='l2', random_state = 8, solver='warn',tol = 0.0001, verbose = 0, warm_start = False.

2: **MNBHyperparameters**→ alpha = 1.0, class-prior = None, fit-prior = True.

3: **SVMHyperparameters**→ *C = 0.1; cache-size = 200; class-weight = None; coef = 0.0; decision-function-shape = 'ovr'; degree = 3;*

```
     Gamma = 'auto_deprecated'; kernel = 'linear'; max-iter = -1; proba-
     bility = True random-state = 8; shrinking = True; tol = 0.001; ver-
     bose = False;
4:   DTHyperparameters→ Class-weight = None Criterion = 'gini'; max-
     depth = None; max-features = None; max-leaf-nodes = None; min-impu-
     rity-decrease = 0.0; min-impurity-split = None; min-samples-
     leaf = 1; min-samples-split = 2; min-weight-fraction-leaf = 0.0;
     presort = False random-state = 0 splitter = 'best';
5:   HaPi(T_input, "classifiername", Hyper - parameters)
```

## Support Vector Machine

The Support Vector Machine (SVM) algorithm is a supervised machine learning approach for categorizing text. It works on the principle of identifying a hyperplane that optimally distinguishes the various groups of people. The term 'Support Vectors' refers to the points close to the hyperplane. If these support vectors are removed, it will cause the hyperplane to shift its location to a different place. The gap between the support vector and the hyperplane is referred to as the margin. It takes a certain number of features for a specific text with a certain label. Pseudocode 2 represents the parameter tuning for the Support Vector Machine-based HaPi (SVM-HaPi) model. After implementing the pseudo-code the confusion matrix is shown in Fig 11.

## Decision tree

Decision trees are data structures that have a topology resembling a tree. To build a tree, training data must first be collected. The tree is then used to generate predictions on test data. The goal of this strategy is to produce the most accurate result feasible while making the fewest number of decisions possible. To tackle problems involving classification and regression, decision trees might be employed. It is a classification technique in which the input space is divided into regions, and each region is assigned a different classification according to the procedure. Pseudocode 2 shows the parameters for Decision Tree based HaPi(DT-HaPi). After implementing the pseudo-code the confusion matrix is shown in Fig 12.

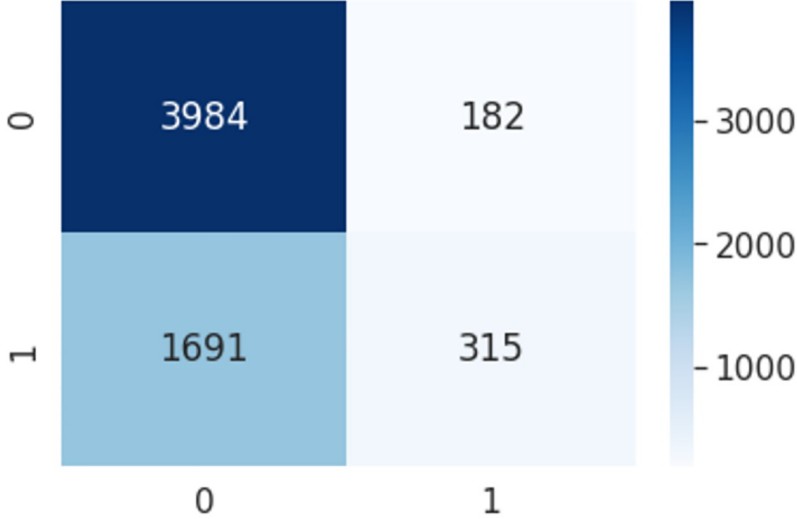

**Fig 11. Confusion metrics using SVM-HaPi algorithm.**

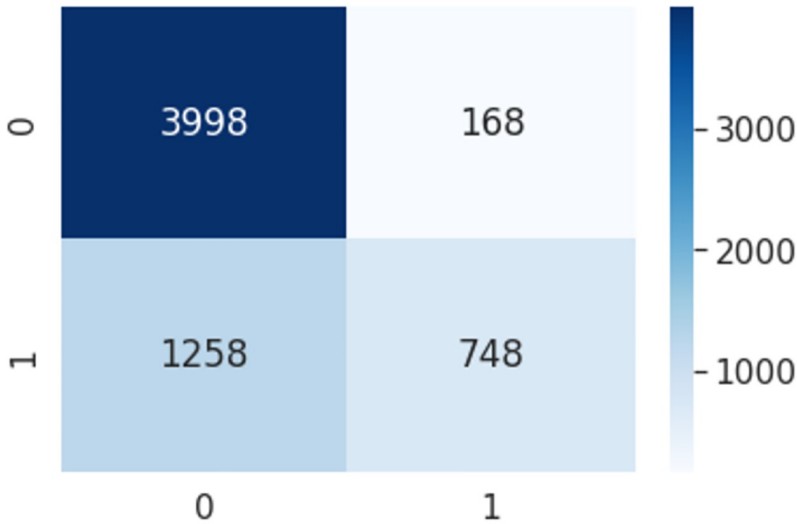

**Fig 12. Confusion metrics using DT-HaPi algorithm.**

The above algorithms are compared based on the evaluation metrics in terms of Precision, Recall, F-measure, and Accuracy. After Comparing all the algorithms it was found that SVM-HaPi showed 69% of Precision, 69% of Recall, 69% of F- Measure, and 69.2% Accuracy which is highest among traditional machine learning algorithms. Table 4 shows the classification report of the Machine Learning algorithms based on the proposed approach.

Fig 13 is the histogram of the comparison of the proposed models in terms of Precision, Recall, F1-score, and Accuracy.

## Validation

The evaluation methods for machine learning models have the issue of being unable to forecast how the model will perform on new data. Cross-validation methods overcome this issue. The goal of Cross-validation is to divide the basic dataset into two parts. The model is first trained on the largest portion of the dataset before being evaluated on the smaller portion. Cross-validation is divided into three categories:

1. **Holdout method**: The dataset is divided into two sets: a training set and a test set. On the training set, the model is fitted. The model is subsequently put to the test on a test set it has never seen before. The mean absolute test error, which is used to evaluate models, is computed using the resulting errors. However, because the variance is usually considerable, the evaluation outcome is heavily dependent on how the test set was picked. As a result, the evaluation outcome can vary greatly between test sets [61].

**Table 4. Classification report based on HaPi.**

| Algorithm | Precison | Recall | F-Measure | Accuracy |
|---|---|---|---|---|
| LR–HaPi | 68% | 67% | 67.4% | 67.8% |
| MN–HaPi | 68% | 68% | 68% | 68% |
| SVM–HaPi | 69% | 69% | 69% | 69.2% |
| DT–HaPi | 68% | 67.5% | 67.7% | 68.5% |

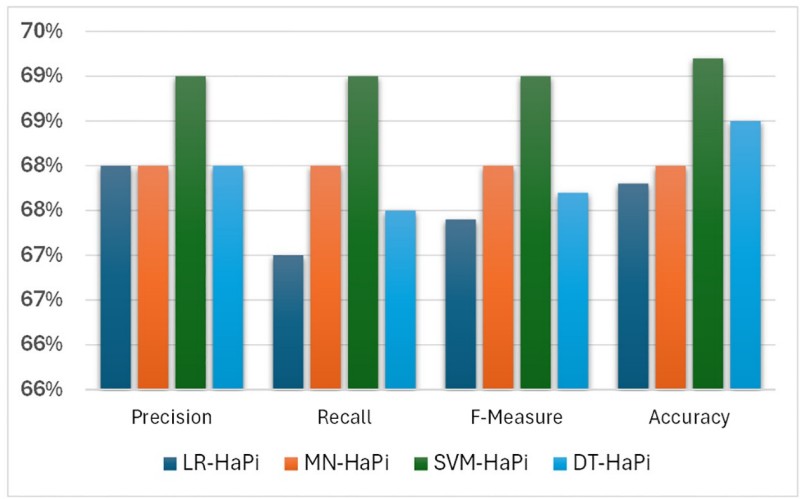

**Fig 13. Performance of different HaPi-based machine learning classifiers.**

2. **k-fold**: The k-fold method can be thought of as a step forward from the holdout approach. The k subsets are chosen, and the holdout method is applied k times, with one of the k subsets serving as a training set and the k-1 subset serving as a test set each time. The average error is then calculated for all k holdout technique runs. The variance decreases as k increases, guaranteeing that the accuracy remains constant across datasets. The disadvantage is that it is more sophisticated and takes longer to run than the holdout approach [62].

3. **Leave-one-out**: The leave-one-out approach is the most extreme version of the k-fold approach, with k equal to the size of the sample universe. Data is trained on all data points except one during each run of the holdout technique, and that one point is then utilized for testing. On the other hand, the computing complexity is high [63].

In order to explore the generalization of our model from training data to unseen data and reduce the possibility of over-fitting, we use five-fold cross-validation. The results showed that there is neither Under-fitting nor over-fitting during the training and testing of the proposed model. The five-fold cross-validation strategy was conducted for all algorithms, and this process was repeated five times. Table 5 reflects the validation mechanism used for checking the reliability and robustness of our proposed approach.

The accuracy of the proposed models on K-Fold cross-validation (k = 5) is shown in Fig 14.

The proposed approach was compared with previous work based on various evaluation metrics. To get a better comparative analysis, we used the same dataset of the previous

**Table 5. Five fold cross validation.**

| Iteration | LR-HaPi | MN-HaPi | SVM-HaPi | DT-HaPi |
|---|---|---|---|---|
| 1 | 67.2% | 67.4% | 68.7% | 68.2% |
| 2 | 68% | 68.5% | 69.2% | 68.8% |
| 3 | 67.9% | 68% | 68.5% | 67.7% |
| 4 | 68.4% | 68.2% | 70.1% | 68% |
| 5 | 67.5% | 67.9% | 69.5% | 69.8% |
| **Mean Accuracy** | **67.8%** | **68%** | **69.2%** | **68.5%** |

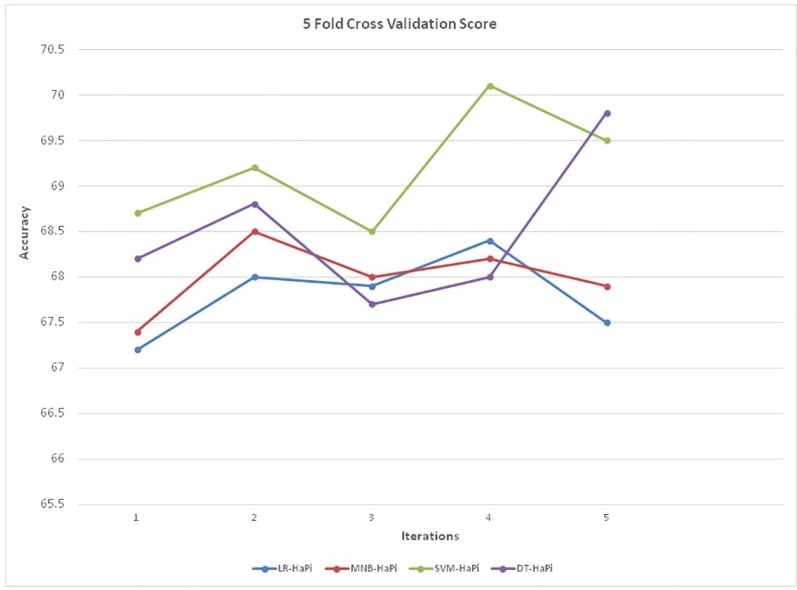

**Fig 14. 5-Fold cross validation.**

researchers which was related to propaganda in our proposed model. The data set that was used for comparison was of QCRI(Qatar Computing Research Institute), Hamad Bin Khalifa University(HBKU), which consists of 536 documents. They labeled the dataset using crowd-sourcing. The machine learning algorithms and other state-of-the-art algorithms like Bidirectional Encoder Representations from Transformers(BERT) were trained and tested on the same dataset using the proposed approach and it was found that our approach outperforms other previously used approaches by giving better results in terms of Precision, Recall, F-Measure, and Accuracy. Among the approaches used SVM-HaPi showed the highest accuracy, by having 64.23% Precision, 65.32 Recall, and F- Measure of 64.77. Our model achieved stable Precision and Recall. Table 6 shows the comparison of Machine Learning algorithms based on the proposed approach with other recent works that are related to Propaganda. To improve the efficiency of the proposed approach more data is needed. Using Deep Neural Networks (DNNs) for propaganda identification in the future is due to their ability to capture complex patterns in textual and visual content, their adaptability to evolving propaganda tactics, and their potential for boosting classification accuracy. Also, these algorithms can automatically learn and extract relevant features from raw data, reducing the need for manual feature engineering and making them adaptable to different domains.

The comparison with previous works is depicted in Fig 15 in terms of various standard evaluation metrics like Precision, Recall, and F-measure.

## Limitations

This study acknowledges potential limitations, with the time frame of data collection being a noteworthy factor. Given the dynamic nature of the propagandistic text, which lacks fixed semantics and undergoes changes over time, there is a risk of overlooking crucial events or shifts in propaganda strategies. Additionally, the effectiveness of the models may diminish when applied to languages, domains, or cultures not adequately represented in the training

**Table 6. Comparison of proposed approaches with the existing studies.**

| Author | Technique | F-Measure | Precison | Recall |
|---|---|---|---|---|
| (Morio et al., 2020) | PoS | 51.55 | 56.54 | 47.37 |
| (Jurkiewicz et al., 2020) | CRF | 49.15 | 59.95 | 41.65 |
| (Chernyavskiy et al., 2020) | Embeddings | 49.10 | 53.23 | 45.56 |
| (Khosla et al., 2020) | Bag of Words | 47.66 | 50.97 | 44.76 |
| (Paraschiv and Cercel, 2020) | Embeddings | 46.6 | 58.61 | 37.94 |
| (Dimov et al., 2020) | n-Grams | 44.68 | 55.62 | 37.34 |
| (Blaschke et al., 2020) | PoS | 43.86 | 42.16 | 45.7 |
| (Verma et al., 2020) | ELMo | 43.60 | 49.86 | 38.74 |
| (Singh et al., 2020) | PoS | 42.21 | 46.52 | 38.63 |
| (Ermurachi and Gifu, 2020) | Bag of Words | 33.21 | 24.49 | 51.57 |
| (Dewantara et al., 2020) | Embeddings | 23.47 | 22.63 | 24.38 |
| (Daval-Frerot & Yannick, 2020) | Embeddings | 18.18 | 34.14 | 12.39 |
| **LR-HaPi** | Hybrid | 61.38 | 63.4 | 59.5 |
| **MNB-HaPi** | Hybrid | 61.37 | 64.3 | 58.7 |
| **SVM-HaPi** | Hybrid | 64.77 | 65.32 | 64.23 |
| **DT-HaPi** | Hybrid | 62.42 | 64.7 | 60.3 |

data. Generalizability becomes a concern, and adaptation may be required for diverse contexts.

The influence of presentation context on whether a text is perceived as propaganda introduces another layer of complexity. Models may encounter challenges in capturing context effectively, leading to instances of false positives or false negatives. Furthermore, the evolving nature of propaganda tactics poses a challenge; a model trained on historical data may struggle to detect contemporary propaganda techniques. Continuous updates and adaptation are imperative to address temporal changes.

To navigate these uncertainties, a cautious and skeptical approach to propaganda identification is essential. Recognizing the limitations related to time frame, linguistic diversity, contextual nuances, and evolving tactics underscores the importance of ongoing model refinement and adaptation.

## Conclusion

In this study, we have proposed a framework for detecting propaganda on social networks. This study will raise public awareness about the impact of propaganda, safeguarding election integrity, and enhancing national security by uncovering disinformation campaigns. Additionally, the proposed framework has potential applications in moderating online platforms, managing crisis responses, protecting brand reputations, and countering medical misinformation during health crises. With its versatility, the system serves as a valuable tool for research, journalism, and legal actions, promoting social cohesion and aiding defense efforts against psychological warfare tactics.

The Twitter public tweets dataset has been used for conducting experiments in this study. Hybrid feature engineering approach in which several features including TF/IDF, Bag of Words, Sentimental features, and Length are merged to train the traditional machine learning algorithms where the SVM-based model (SVM-HaPi) exhibits superior performance among traditional machine learning algorithms by achieving a precision, recall, F-Measure, and overall accuracy of 69.0%, 69.0%, 69.0%, and 69.2%, respectively.

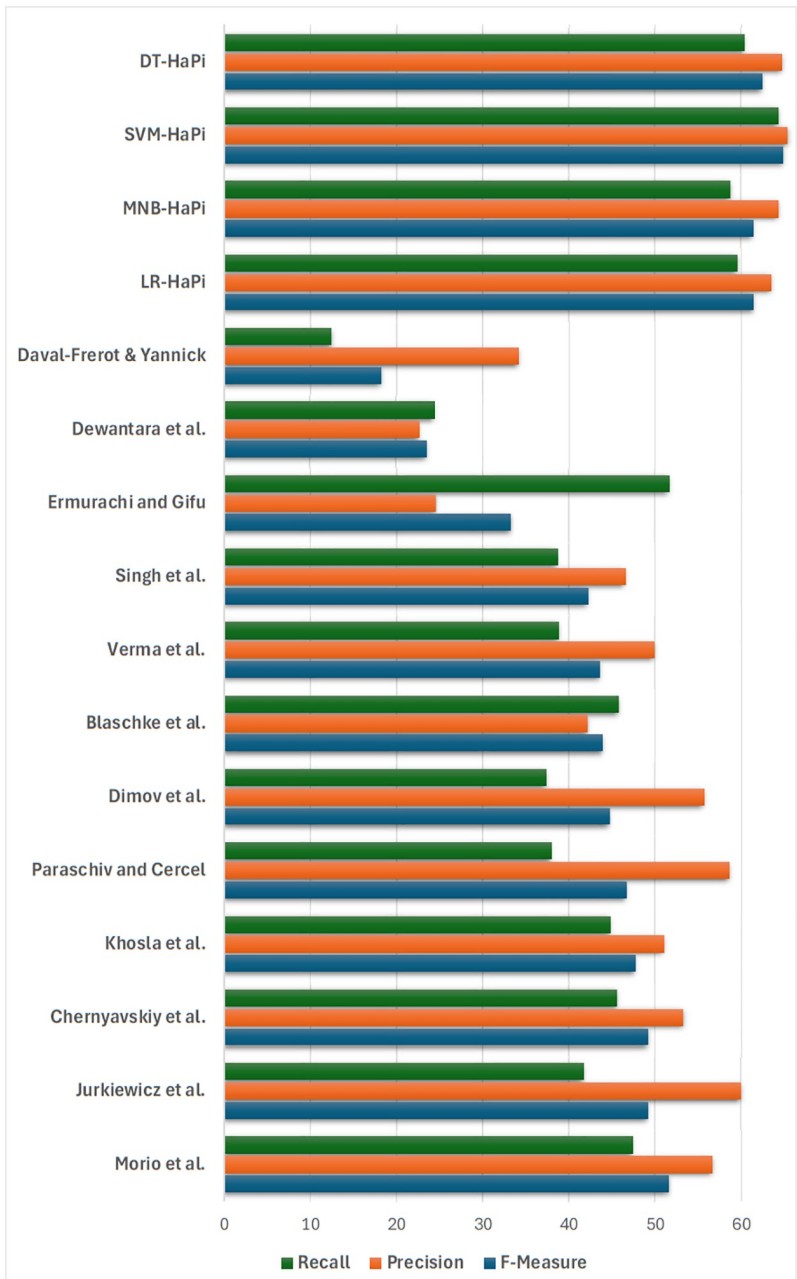

**Fig 15. Comparison of the proposed approach with existing studies.**

The scope of propaganda identification extends beyond textual data alone, as Deep Neural Networks (DNNs) can be expanded to handle multimodal data encompassing text, images, audio, and video. By integrating both textual and visual features, propaganda detection can be enhanced to consider not just the content, but also the presentation and context in which it is disseminated. Furthermore, the proposed framework has the potential to evolve into a comprehensive system incorporating various modules, including detecting misinformation, identifying fake content and profiles, and determining the stance using social media data.

Furthermore, looking forward, our future research endeavors aim to explore ensemble methods to augment the reliability and predictive accuracy of our studies. Through an exploration of ensemble techniques' synergies, we aim to gain deeper insights, refine our models, and contribute significantly to the ongoing advancement of our research domain. Additionally, while our current study focuses solely on English text, forthcoming research directions may entail the exploration of multilingual corpora, integrating datasets in languages such as Hindi, Urdu, and German, which are now publicly accessible.

## Acknowledgments

This work is supported by EIAS (Emerging Intelligent Autonomous Systems) Data Science Lab, Prince Sultan University, KSA). The authors would like to acknowledge the support of Prince Sultan University for paying the Article Processing Charges (APC) of this publication.

## Author Contributions

**Conceptualization:** Akib Mohi Ud Din Khanday, Mudasir Ahmad Wani.

**Data curation:** Akib Mohi Ud Din Khanday, Mudasir Ahmad Wani.

**Formal analysis:** Akib Mohi Ud Din Khanday, Mudasir Ahmad Wani, Qamar Rayees Khan.

**Funding acquisition:** Mudasir Ahmad Wani, Ahmed A. Abd El-Latif.

**Investigation:** Syed Tanzeel Rabani.

**Methodology:** Mudasir Ahmad Wani.

**Project administration:** Syed Tanzeel Rabani, Qamar Rayees Khan, Ahmed A. Abd El-Latif.

**Resources:** Mudasir Ahmad Wani.

**Supervision:** Mudasir Ahmad Wani, Qamar Rayees Khan, Ahmed A. Abd El-Latif.

**Validation:** Mudasir Ahmad Wani, Syed Tanzeel Rabani, Qamar Rayees Khan, Ahmed A. Abd El-Latif.

**Visualization:** Mudasir Ahmad Wani.

**Writing – original draft:** Akib Mohi Ud Din Khanday, Mudasir Ahmad Wani.

**Writing – review & editing:** Akib Mohi Ud Din Khanday, Syed Tanzeel Rabani, Qamar Rayees Khan, Ahmed A. Abd El-Latif.

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
