## [Decision Letter · Decision Letter 0]

5 Mar 2024

PONE-D-24-02023HAPI: An efficient Hybrid Feature Engineering-based Framework for Propaganda Identification in Social MediaPLOS ONE

Dear Dr. Wani,

Thank you for submitting your manuscript to PLOS ONE. After careful consideration, we feel that it has merit but does not fully meet PLOS ONE’s publication criteria as it currently stands. Therefore, we invite you to submit a revised version of the manuscript that addresses the points raised during the review process.

We look forward to receiving your revised manuscript.

Kind regards,

Josep Vidal-Alaball, MD, PdD, MPH

Academic Editor

PLOS ONE

2. (1) In your Methods section, please include additional information about your dataset and ensure that you have included a statement specifying whether the collection and analysis method complied with the terms and conditions for the source of the data.

(2) Please note that PLOS ONE has specific guidelines on code sharing for submissions in which author-generated code underpins the findings in the manuscript. In these cases, all author-generated code must be made available without restrictions upon publication of the work. Please review our guidelines at https://journals.plos.org/plosone/s/materials-and-software-sharing#loc-sharing-code and ensure that your code is shared in a way that follows best practice and facilitates reproducibility and reuse.

3. In this instance it seems there may be acceptable restrictions in place that prevent the public sharing of your minimal data. However, in line with our goal of ensuring long-term data availability to all interested researchers, PLOS’ Data Policy states that authors cannot be the sole named individuals responsible for ensuring data access (http://journals.plos.org/plosone/s/data-availability#loc-acceptable-data-sharing-methods).

4. We note that Figures 6 and 7 in your submission contain copyrighted images. All PLOS content is published under the Creative Commons Attribution License (CC BY 4.0), which means that the manuscript, images, and Supporting Information files will be freely available online, and any third party is permitted to access, download, copy, distribute, and use these materials in any way, even commercially, with proper attribution. For more information, see our copyright guidelines: http://journals.plos.org/plosone/s/licenses-and-copyright.

1. You may seek permission from the original copyright holder of Figures 6 and 7 to publish the content specifically under the CC BY 4.0 license.

Additional Editor Comments:

Following the reviewers' feedback, we kindly ask you to revise your manuscript.

Reviewers' comments:

Reviewer's Responses to Questions

**Comments to the Author**

1. Is the manuscript technically sound, and do the data support the conclusions?

Reviewer #1: Yes

Reviewer #2: Yes

2. Has the statistical analysis been performed appropriately and rigorously? 

Reviewer #1: Yes

Reviewer #2: Yes

3. Have the authors made all data underlying the findings in their manuscript fully available?

Reviewer #1: No

Reviewer #2: No

4. Is the manuscript presented in an intelligible fashion and written in standard English?

Reviewer #1: Yes

Reviewer #2: Yes

5. Review Comments to the Author

Reviewer #1: Thank you for the kind opportunity to review this manuscript, I read it with great interest. The study is conducted on a hot topic and develops a framework for propaganda identification in social media. The study introduces a novel hybrid feature selection methodology that leverages the strengths of various techniques to improve the accuracy and reliability of propaganda identification systems. There are several strengths of the paper such as good reference to previous work, good use of figures, and generally well written.

I have very minor comments:

• Check acronyms are written in full the first time, e.g. ‘OSNs’

• Check size of some visuals e.g., figure 2, figure 5, figure 13

• Check word cloud construction (figure 5 and 7) – can these be enlarged or their colour changes, not all words were visible.

• Could the formulas/mathematical notations be embedded better within the paper e.g. page 17, lines 533 to 547 and some textual description (short sentence or two explaining what they mean) could help improve it.

• The formula/mathematical notions described above could be explained/connected to Figure 2/Figure 1 to contextualise where they fall.

• There is some repetition in the conclusion from the abstract and paper itself, the conclusion could be improved e.g. potential use cases.

Reviewer #2: The paper titled "HAPI: An Efficient Hybrid Feature Engineering-based Framework for Propaganda Identification in Social Media" introduces HAPI, a novel Hybrid Feature Engineering Approach designed for identifying propaganda in text-based content. This approach integrates traditional feature engineering techniques with machine learning methods, utilizing Twitter data and a binary annotation scheme. Overall, the paper presents a commendable contribution to this domain and maintains a satisfactory level of quality. However, I have several concerns and feedback as under:

• The title "HAPI: An efficient Hybrid Feature Engineering-based Framework for Propaganda Identification in Social Media" lacks clarity regarding the meaning of 'A' in the abbreviation. I recommend that the authors revise the title accordingly for clarity.

• It is advisable to define and elaborate on all abbreviations and acronyms upon their first appearance in the paper.

• The abstract appears rushed and lacks sufficient detail, particularly regarding comparisons and future research directions. Enhancing these aspects would improve the quality of the abstract.

• The literature review section would benefit from a comprehensive concluding paragraph that summarizes the key points drawn from the literature in a clear manner.

• It is recommended to provide a definition of propaganda and its synonyms at the beginning of the introduction to facilitate reader understanding.

• The authors are required to furnish a detailed elucidation of the word clouds depicted in figures 6 and 7. Furthermore, if feasible, they should offer exemplar sentences corresponding to select words within both categories derived from the amassed dataset.

• Figure 2, depicting the flowchart of processes, suffers from poor visibility, and the caption requires clarification. The authors are urged to improve the figure's quality and update the caption accordingly in the revised version.

• Figure 3, illustrating the framework for data extraction from Twitter, appears stretched and requires correction.

• A pictorial representation of the Data Preprocessing (section 2.2) would enhance the clarity and understanding of the process.

• The title of section 3, "Results of Propaganda Identification based on HaPi," lacks clarity and should be revised for better coherence.

• Algorithm 1, detailing the HAPI approach, appears improperly formatted. The authors are encouraged to present it in a well-defined format suitable for journal standards.

• Several section numbers are incorrect throughout the manuscript and should be updated accordingly.

• Figure 12, depicting the Comparison of Machine Learning Classifiers based on HAPI, suffers from blurriness, and requires enhancement for improved clarity.

Overall, while the paper presents valuable insights into propaganda identification in social media, addressing the concerns and incorporating the suggested improvements would significantly enhance its quality and readability.

• I further suggest removing references to the arxiv papers or replace them with their peer-reviewed versions.

6. PLOS authors have the option to publish the peer review history of their article (what does this mean?). If published, this will include your full peer review and any attached files.

Reviewer #1: No

Reviewer #2: No

---

## [Author Response · Author response to Decision Letter 0]

3 Apr 2024

Editorial, Requirement # 1 Please ensure that your manuscript meets PLOS ONE's style requirements, including those for file naming.

Author response: We followed the PLOS ONE’s latex template for the style requirements and all the files are renamed as mentioned in the guidelines. ________________________________________

Editorial, Requirement # 2 (1) In your Methods section, please include additional information about your dataset and ensure that you have included a statement specifying whether the collection and analysis method complied with the terms and conditions for the source of the data.

Author Actions:

We have now included the dataset statistics and data collection process in detail including whether collection and analysis methods obey the terms and conditions specified by the data source in the revised manuscript. 

The newly added content is highlighted in the revised manuscript.

(2) Please note that PLOS ONE has specific guidelines on code sharing for submissions in which author-generated code underpins the findings in the manuscript. In these cases, all author-generated code must be made available without restrictions upon publication of the work

Author Actions:

In compliance with the code-sharing policy of PLOS ONE, we ensured that all author-generated code used to support the findings presented in our manuscript is made openly accessible on our GitHub account. The link to the repository is provided in the Data availability statement. This will include providing access to the code through a publicly accessible git repository. We have taken the necessary steps to prepare the code for sharing and have included details on how to access it in the revised manuscript. 

Editor, Requirement# 3 : In this instance, it seems there may be acceptable restrictions in place that prevent the public sharing of your minimal data. However, in line with our goal of ensuring long-term data availability to all interested researchers, PLOS’ Data Policy states that authors cannot be the sole named individuals responsible for ensuring data access (http://journals.plos.org/plosone/s/data-availability#loc-acceptable-data-sharing-methods).

Authors' Response: We sincerely appreciate your thorough review of our manuscript and your insightful comments regarding data accessibility by PLOS' Data Policy. 

In response to your valuable feedback, we have taken the necessary steps to designate a non-author contact who possesses the requisite administrative access to our dataset. She has extensive experience and expertise in the field and is well-suited to handle data access requests from fellow researchers and ensure compliance with data-sharing policies.

You may find Dr. Shakil's contact information below for your reference:

Name: Dr. Kashish Ara Shakil

Designation: Associate Professor

Department: College of Computer and Information Sciences

Institution: Princess Nora bint Abdulrahman University

Country: Riyadh, KSA

Phone: +966 0545009560

Email: kashakil@pnu.edu.sa

We assure you that Dr. Shakil is fully committed to upholding the principles of data accessibility and will diligently oversee the process of responding to requests, thereby ensuring the persistent and long-term availability of our dataset.

Editor, Requirement# 4: We note that Figures 6 and 7 in your submission contain copyrighted images. All PLOS content is published under the Creative Commons Attribution License (CC BY 4.0), which means that the manuscript, images, and Supporting Information files will be freely available online, and any third party is permitted to access, download, copy, distribute, and use these materials in any way, even commercially, with proper attribution.

Author response: We sincerely appreciate your thorough review of our manuscript, as per the suggestion we have updated the word clouds with different colors and shape.

Author action: We updated the manuscript by modifying the Figure 6 and 7 with different shapes and colors. Although the previous version had no copyright issues as we draw that using python library, in the modified version we changed the shape and colors in order to be it more visible.

Reviewer#1, Concern # 1 (Check acronyms are written in full the first time e.g., OSNs): 

Author response: Thank you for assessing this, we have checked all the acronyms and have modified the manuscript by adding their full forms in their first appearance.

Author action: We updated the manuscript by providing the full form of acronyms when mentioned in the first appearance.

Reviewer#1, Concern # 2 (Check Size of some Visuals e.g. figure 2, figure 5 and figure 13): 

Author response: We appreciate your assessment that figures 2, 5, and 13 are modified.

Author action: We updated the manuscript by modifying the figures 2,5 and 13.

Reviewer#1, Concern # 3 (Check word cloud construction (Figure 5 and 7) can these be enlarged or their colour change not all words were visible)

Author response: Thank you for pointing this out, as per the suggestion we have updated the word clouds with different colors and shapes.

Author action: We updated the manuscript by modifying Figures 6 and 7 with different shapes and colors. Although the previous version had no copyright issues as we drew that using the Python library, in the modified version we changed the shape and colors in order to make it more visible.

Reviewer#1, Concern # 4 (There is some repetition in the conclusion from the abstract and paper itself, the conclusion could be improved e.g. potential use cases)

Author response: Thank you for your valuable feedback. We acknowledge the repetition in the conclusion and agree that enhancing the conclusion with potential use cases would improve the overall clarity and impact of the paper. 

Author Action: In response to the reviewer's feedback, we have revised the manuscript by eliminating repetitions in both the conclusion and abstract sections. Additionally, we have included new content to enhance the details provided in these sections. The new updated and newly added content is highlighted in the revised manuscript. 

We appreciate the valuable suggestion and have taken steps to address it in the revised manuscript.________________________________________

Reviewer#2, Concern # 1 (The title "HAPI: An efficient Hybrid Feature Engineering-based Framework for Propaganda Identification in Social Media" lacks clarity regarding the meaning of 'A' in the abbreviation. I recommend that the authors revise the title accordingly for clarity.)

Author response: Thank you for pointing this out, we have modified the title to “HAPI: An efficient Hybrid Feature Engineering-based Approach for Propaganda Identification in Social Media.”

Author action: We updated the manuscript by modifying the title to “HAPI: An efficient Hybrid Feature Engineering-based Approach for Propaganda Identification in Social Media” such that the meaning of ‘A’ in the abbreviation is clear.

Reviewer#2, Concern # 2 (It is advisable to define and elaborate on all abbreviations and acronyms upon their first appearance in the paper.)

Author response: We appreciate your assessment and we have checked all the acronyms and elaborated on their first appearance.

Author action: We updated the manuscript by elaborating all the acronyms in their first appearance.

Reviewer#2, Concern # 3 (The abstract appears rushed and lacks sufficient detail, particularly regarding comparisons and future research directions. Enhancing these aspects would improve the quality of the abstract.)

Author response: Thank you for your feedback on the abstract. We appreciate your insights regarding the need for more detail, particularly in terms of comparisons and future research directions. 

Author action: As per the reviewer's suggestions we have now revised the abstract to provide a more comprehensive overview of the study, including clearer comparisons with existing research and a discussion of potential future research directions. 

Reviewer#2, Concern # 4 (The literature review section would benefit from a comprehensive concluding paragraph that summarizes the key points drawn from the literature in a clear manner.)

Author response: We appreciate your assessment and we have added the concluding paragraph that summarizes the key findings from the literature.

Author action: We updated the manuscript by adding a concluding paragraph “News article analysis has been the main focus of most research in this area. On the other hand people in the modern era share their thoughts and viewpoints on social media platforms, and there is a great potential to use online social media data for propaganda detection. These platforms are excellent resources for current events and provide insightful information about the propagation of propaganda and its impact on public opinion. The performance of several machine learning classifiers in this context has not been sufficiently enhanced by feature selection techniques. By determining the most relevant and discriminative features in the dataset, feature selection will play a critical role in managing the effectiveness of classifiers.”

Reviewer#2, Concern # 5 (It is recommended to provide a definition of propaganda and its synonyms at the beginning of the introduction to facilitate reader understanding.)

Author response: Thank you for pointing this out, we have added the definition of propaganda in the beginning of the introduction.

Author action: We updated the manuscript by adding a definition of propaganda at start of the introduction “Propaganda refers to the dissemination of biased or misleading information in order to influence the opinions, beliefs, attitudes, or behaviors of individuals or groups in a particular direction”

Reviewer#2, Concern # 6 (The authors are required to furnish a detailed elucidation of the word clouds depicted in figures 6 and 7. Furthermore, if feasible, they should offer exemplar sentences corresponding to select words within both categories derived from the amassed dataset.)

Author response: Thank you for pointing this out, as per the suggestion we have updated the word clouds with different colors and shapes.

Author action: We updated the manuscript by modifying Figures 6 and 7 with different shapes and colors. Although the previous version had no copyright issues as we drew that using the python library, in the modified version we changed the shape and colors in order to make it more visible. Also, the words that are more frequent are mentioned in the revised manuscript “Upon close examination, it is evident that posts with propagandistic content tend to revolve on particular subjects, such as politics, religion, trending events, and influential people. This finding is corroborated by a detailed examination of the word cloud linked to propaganda, which shows that terms pertaining to politics, religion, ISIS, and Islam appear most frequently. This implies a deliberate attempt to advance specific ideologies or agendas by disseminating inaccurate or biased information on a variety of digital platforms.”.

Reviewer#2, Concern # 7 (Figure 2, depicting the flowchart of processes, suffers from poor visibility, and the caption requires clarification. The authors are urged to improve the figure's quality and update the caption accordingly in the revised version.)

Author response: We have modified Figure 2 and updated its caption for better clarification.

Author action: We updated the revised manuscript by modifying Figure 2.

Reviewer#2, Concern # 8 (Figure 3, illustrating the framework for data extraction from Twitter, appears stretched and requires correction.)

Author response: We appreciate your assessment and we have checked Figure 3 and modified its dimensions.

Author action: We updated the manuscript by modifying the size of Figure 3.

Reviewer#2, Concern # 9(A pictorial representation of the Data Preprocessing (section 2.2) would enhance the clarity and understanding of the process.)

Author response: Thanks for the suggestion, as per the suggestion we have added the pictorial representation of the Data Preprocessing section.

Author action: We updated the manuscript by adding figure 8, which depicts the major steps in data preprocessing.

Reviewer#2, Concern # 10 (The title of section 3, "Results of Propaganda Identification based on HaPi," lacks clarity and should be revised for better coherence.)

Author response: We appreciate your assessment and we have modified the title of section 3.

Author action: We updated the manuscript by changing the title of section 3 to “Results and Discussion”.

Reviewer#2, Concern # 11 (Algorithm 1, detailing the HAPI approach, appears improperly formatted. The authors are encouraged to present it in a well-defined format suitable for journal standards.)

Author response: We have rectified the formatting issues in the Algorithm 1.

Author action: We updated the manuscript by modifying the Algorithm 1.

Reviewer#2, Concern # 12 (Several section numbers are incorrect throughout the manuscript and should be updated accordingly.)

Author response: We appreciate your assessment and we have checked the section numbering and rectified the errors in numbering.

Author action: We updated the manuscript by following the numbering style in each section.

Reviewer#2, Concern # 13 ( Figure 12, depicting the Comparison of Machine Learning Classifiers based on HAPI, suffers from blurriness, and requires enhancement for improved clarity.

Overall, while the paper presents valuable insights into propaganda identification in social media, addressing the concerns and incorporating the suggested improvements would significantly enhance its quality and readability.)

Author response: Thank you for your valuable feedback. We acknowledge the concern regarding the clarity of Figure 12 and recognize the importance of enhancing its quality for improved readability. We have modified Figure 12 and its resolution.

Author action: We updated the manuscript by modifying Figure 12 to look more clear, readable and visible. 

Reviewer#2, Concern # 14 (I further suggest removing references to the arxiv papers or replace them with their peer-reviewed versions.)

Author response: We appreciate your assessment and we have checked the arxiv papers and replaced them with their peer review, if available. 

Author action: We updated the manuscript by modifying the references of arxiv papers with their peer-reviewed ve

---

## [Editor Report · Decision Letter 1]

9 Apr 2024

HAPI: An efficient Hybrid Feature Engineering-based Approach for Propaganda Identification in Social Media

PONE-D-24-02023R1

Dear Dr. Wani,

We’re pleased to inform you that your manuscript has been judged scientifically suitable for publication and will be formally accepted for publication once it meets all outstanding technical requirements.

Kind regards,

Josep Vidal-Alaball, MD, PdD, MPH

Academic Editor

PLOS ONE
---

## [Editor Report · Acceptance letter]

30 May 2024

PONE-D-24-02023R1 

PLOS ONE

Dear Dr. Wani, 

I'm pleased to inform you that your manuscript has been deemed suitable for publication in PLOS ONE. Congratulations! Your manuscript is now being handed over to our production team.

Kind regards, 

on behalf of

Dr. Josep Vidal-Alaball 

Academic Editor

PLOS ONE